

# De novo clustering methods outperform reference-based methods for assigning 16S rRNA gene sequences to operational taxonomic units

Sarah L. Westcott and Patrick D. Schloss

Department of Microbiology and Immunology, University of Michigan—Ann Arbor, Ann Arbor, MI, United States

Corresponding author
Patrick D. Schloss,
pschloss@umich.edu

## ABSTRACT

**Background.** 16S rRNA gene sequences are routinely assigned to operational taxonomic units (OTUs) that are then used to analyze complex microbial communities. A number of methods have been employed to carry out the assignment of 16S rRNA gene sequences to OTUs leading to confusion over which method is optimal. A recent study suggested that a clustering method should be selected based on its ability to generate stable OTU assignments that do not change as additional sequences are added to the dataset. In contrast, we contend that the quality of the OTU assignments, the ability of the method to properly represent the distances between the sequences, is more important.

**Methods.** Our analysis implemented six *de novo* clustering algorithms including the single linkage, complete linkage, average linkage, abundance-based greedy clustering, distance-based greedy clustering, and Swarm and the open and closed-reference methods. Using two previously published datasets we used the Matthew's Correlation Coefficient (MCC) to assess the stability and quality of OTU assignments.

**Results.** The stability of OTU assignments did not reflect the quality of the assignments. Depending on the dataset being analyzed, the average linkage and the distance and abundance-based greedy clustering methods generated OTUs that were more likely to represent the actual distances between sequences than the open and closed-reference methods. We also demonstrated that for the greedy algorithms VSEARCH produced assignments that were comparable to those produced by USEARCH making VSEARCH a viable free and open source alternative to USEARCH. Further interrogation of the reference-based methods indicated that when USEARCH or VSEARCH were used to identify the closest reference, the OTU assignments were sensitive to the order of the reference sequences because the reference sequences can be identical over the region being considered. More troubling was the observation that while both USEARCH and VSEARCH have a high level of sensitivity to detect reference sequences, the specificity of those matches was poor relative to the true best match.

**Discussion.** Our analysis calls into question the quality and stability of OTU assignments generated by the open and closed-reference methods as implemented in current version of QIIME. This study demonstrates that *de novo* methods are the optimal method of assigning sequences into OTUs and that the quality of these

assignments needs to be assessed for multiple methods to identify the optimal clustering method for a particular dataset.

## INTRODUCTION

The ability to affordably generate millions of 16S rRNA gene sequences has allowed microbial ecologists to thoroughly characterize the microbial community composition of hundreds of samples. To simplify the complexity of these large datasets, it is helpful to cluster sequences into meaningful bins. These bins, commonly known as operational taxonomic units (OTUs), are used to compare the biodiversity contained within and between different samples (*Schloss & Westcott, 2011*). Such comparisons have enabled researchers to characterize the microbiota associated with the human body (e.g., *Huttenhower et al., 2012*), soil (e.g., *Shade et al., 2013*), aquatic ecosystems (e.g., *Gilbert et al., 2011*), and numerous other environments. Within the field of microbial ecology, a convention has emerged where sequences are clustered into OTUs using a threshold of 97% similarity or a distance of 3%. One advantage of the OTU-based approach is that the definition of the bins is operational and can be changed to suit the needs of the particular project. However, with the dissemination of clustering methods within software such as mothur (*Schloss et al., 2009*), QIIME (*Caporaso et al., 2010*), and other tools (*Sun et al., 2009*; *Edgar, 2010*; *Edgar, 2013*; *Cai & Sun, 2011*; *Mahé et al., 2014*), it is important to understand how different clustering methods implement this conventional OTU threshold. Furthermore, it is necessary to understand how the selected method affects the precision and accuracy of assigning sequences to OTUs. Broadly speaking, three approaches have been developed to assign sequences to OTUs.

The first approach has been referred to as phylotyping (*Schloss & Westcott, 2011*) or closed-reference clustering (*Navas-Molina et al., 2013*). This approach involves comparing sequences to a curated database and then clustering sequences into the same OTU that are similar to the same reference sequence. Reference-based clustering methods suffer when the reference does not adequately reflect the biodiversity of the community. If a large fraction of sequences are novel, then they cannot be assigned to an OTU. In addition, the reference sequences are selected because they are less than 97% similar to each other over the full length of the gene; however, it is known that the commonly used variable regions within the 16S rRNA gene do not evolve at the same rate as the full-length gene (*Schloss, 2010*; *Kim, Morrison & Yu, 2011*). Thus, a sequence representing a fragment of the gene may be more than 97% similar to multiple reference sequences. Defining OTUs in the closed-reference approach is problematic because two sequences might be 97% similar to the same reference sequence, but they may only be 94% similar to each other. Alternatively, a sequence may be equally similar to two or more reference sequences. An

alternative to this approach is to use a classifier to assign a taxonomy to each sequence so that sequences can be clustered at a desired level within the Linnean taxonomic hierarchy (*Schloss & Westcott, 2011*). The strengths of the reference-based methods include their speed, potential for trivial parallelization, ability to compare OTU assignments across studies, and the hope that as databases improve, the OTU assignments will also improve.

The second approach has been referred to as distance-based (*Schloss & Westcott, 2011*) or *de novo* clustering (*Navas-Molina et al., 2013*). In this approach, the distance between sequences is used to cluster sequences into OTUs rather than the distance to a reference database. In contrast to the efficiency of closed-reference clustering, the computational cost of hierarchical *de novo* clustering methods scales quadratically with the number of unique sequences. The expansion in sequencing throughput combined with sequencing errors inflates the number of unique sequences resulting in the need for large amounts of memory and time to cluster the sequences. If error rates can be reduced through stringent quality control measures, then these problems can be overcome (*Kozich et al., 2013*). As an alternative, heuristics have been developed to approximate the clustering of hierarchical methods (*Sun et al., 2009*; *Edgar, 2010*; *Mahé et al., 2014*). Two related heuristics implemented in USEARCH were recently described: distance-based greedy clustering (DGC) and abundance-based greedy clustering (AGC) (*Edgar, 2010*; *He et al., 2015*). These greedy methods cluster sequences within a defined similarity threshold of an index sequence or create a new index sequence. If a sequence is more similar than the defined threshold, it is assigned to the closest centroid based (i.e., DGC) or the most abundant centroid (i.e., AGC). One critique of *de novo* approaches is that OTU assignments are sensitive to the input order of the sequences (*Mahé et al., 2014*; *He et al., 2015*). Whether the differences in assignments is meaningful is unclear and the variation in results could represent equally valid clustering of the data. The strength of *de novo* clustering is its independence of references for carrying out the clustering step. For this reason, *de novo* clustering has been preferred across the field. After clustering, the classification of each sequence can be used to obtain a consensus classification for the OTU (*Schloss & Westcott, 2011*).

The third approach, open-reference clustering, is a hybrid of the closed-reference and *de novo* approaches (*Navas-Molina et al., 2013*; *Rideout et al., 2014*). Open-reference clustering involves performing closed-reference clustering followed by *de novo* clustering on those sequences that are not sufficiently similar to the reference. In theory, this method should exploit the strengths of both closed-reference and *de novo* clustering; however, the different OTU definitions employed by commonly used closed-reference and *de novo* clustering implementations pose a possible problem when the methods are combined. An alternative to this approach has been to classify sequences to a bacterial family or genus and then assign those sequences to OTUs within those taxonomic groups using the average linkage method (*Schloss & Westcott, 2011*). For example, all sequences classified as belonging to the *Porphyromonadaceae* would then be assigned to OTUs using the average linkage method using a 3% distance threshold. Those sequences that did not classify to a known family would also be clustered using the average linkage method. An advantage

of this approach is that it lends itself nicely to parallelization since each taxonomic group is seen as being independent and can be processed separately. Such an approach would overcome the difficulty of mixing OTU definitions between the closed-reference and *de novo* approaches; however, it would still suffer from the problems associated with database quality and classification error.

The growth in options for assigning sequences using each of these three broad approaches has been considerable. It has been difficult to objectively assess the quality of OTU assignments. Some have focused on the time and memory required to process a dataset (*Sun et al., 2009*; *Cai & Sun, 2011*; *Mahé et al., 2014*; *Rideout et al., 2014*). These are valid parameters to assess when judging a clustering method, but have little to say about the quality of the OTU assignments. Others have attempted to judge the quality of a method by its ability to generate data that parallels classification data (*White et al., 2010*; *Sun et al., 2011*; *Cai & Sun, 2011*). This approach is problematic because bacterial taxonomy often reflects historical biases amongst bacterial systematicists. Furthermore, it is well known that the rates of evolution across lineages are not the same (*Wang et al., 2007*; *Schloss, 2010*). A related approach has used clustering of mock community data to evaluate methods (*Huse et al., 2010*; *Barriuso, Valverde & Mellado, 2011*; *Bonder et al., 2012*; *Chen et al., 2013*; *Edgar, 2013*; *Mahé et al., 2014*; *May et al., 2014*). Yet these approaches ignore the effects of sequencing errors that tend to accumulate with sequencing depth and represent highly idealized communities that lack the phylogenetic diversity of real microbial communities (*Schloss, Gevers & Westcott, 2011*; *Kozich et al., 2013*). Others have assessed the quality of clustering based on their ability to generate the same OTUs generated by other methods (*Rideout et al., 2014*; *Schmidt, Rodrigues & Mering, 2014b*). This is problematic because it does not solve the fundamental question of which method is optimal. The concept of ecological consistency as a metric of quality asserts that sequences that cluster into the same OTU should share similar ecological affiliations (*Koeppel & Wu, 2013*; *Preheim et al., 2013*; *Schmidt, Rodrigues & Mering, 2014a*). Although this is an intriguing approach and is a quantitative metric, it is unclear how the metric would be objectively validated. We recently proposed an approach for evaluating OTU assignments using the distances between pairs of sequences (*Schloss & Westcott, 2011*). We were able to synthesize the relationship between OTU assignments and the distances between sequences using the Matthew's correlation coefficient (MCC; *Matthews, 1975*). MCC can be interpreted as representing the correlation between the observed and expected classifications and can vary between $-1.0$ and $1.0$. The strength of the MCC, as implemented by *Schloss & Westcott (2011)*, is that it is an objective approach to assessing the quality of the OTU assignments that can be calculated for any set of OTU assignments where there is a distance matrix and a specific threshold without relying on an external reference.

A recent analysis by He and colleagues (*2015*) characterized the three general clustering approaches by focusing on what they called stability. They defined stability as the ability of a method to provide the same clustering on a subset of the data as was found in the full dataset. Their concept of stability did not account for the quality of the OTU assignments and instead focused on the precision of the assignments. A method may be very stable, but

of poor quality. In the current analysis, we assessed the quality and stability of the various clustering methods. Building on our previous analysis of clustering methods, our hypothesis was that the methods praised by the He study for their stability actually suffered a lack of quality. In addition, we assess these parameters in light of sequence quality using the original 454 dataset and a larger and more modern dataset generated using the MiSeq platform.

## METHODS

### 454 FLX-generated Roesch Canadian soil dataset

After obtaining the 16S rRNA gene fragments from GenBank (accessions EF308591–EF361836), we followed the methods outlined by the He study by removing any sequence that contained an ambiguous base, was identified as being a chimera, and fell outside a defined sequence length. Although they reported observing a total of 50,542 sequences that were represented by 13,293 unique sequences, we obtained a total of 50,946 sequences that were represented by 13,393 unique sequences. Similar to the He study, we randomly sampled, without replacement, 20, 40, 60, and 80% of the sequences from the full data set. The random sampling was repeated 30 times. The order of the sequences in the full dataset was randomly permuted without replacement to generate an additional 30 datasets. To perform the hierarchical clustering methods and to generate a distance matrix we followed the approach of the He study by calculating distances based on pairwise global alignments using the `pairwise.dist` command in mothur using the default Needleman-Wunsch alignment method and parameters. It should be noted that this approach has been strongly discouraged (*Schloss, 2012*). Execution of the hierarchical clustering methods was performed as described in the original He study using mothur (v.1.37) and using the QIIME (v.1.9.1) parameter profiles provided in the supplementary material from the He study for the greedy and reference-based clustering methods.

### MiSeq-generated Murine gut microbiota dataset

The murine 16S rRNA gene sequence data generated from the V4 region using an Illumina MiSeq was obtained from http:/www.mothur.org/MiSeqDevelopmentData/StabilityNoMetaG.tar  and was processed as outlined in the original study (*Kozich et al., 2013*). Briefly, 250-nt read pairs were assembled into contigs by aligning the reads and correcting discordant base calls by requiring one of the base calls to have a Phred quality score at least 6 points higher than the other. Sequences where it was not possible to resolve the disagreement were culled from the dataset. The sequences were then aligned to a SILVA reference alignment (*Pruesse et al., 2007*) and any reads that aligned outside of the V4 region were removed from the dataset. Sequences were pre-clustered by combining the abundances of sequences that differed by 2 or fewer nucleotides of a more abundant sequence. Each of the samples was then screened for chimeric sequences using the default parameters in UCHIME (*Edgar et al., 2011*). The resulting sequences were processed in the same manner as the Canadian soil dataset with the exception that the distance matrices were calculated based on the SILVA-based alignment.

### Analysis of reference database

We utilized the 97% OTUs greengenes reference sequence and taxonomy data (v.13.8) that accompanies the QIIME installation. Because the greengenes reference alignment does a poor job of representing the secondary structure of the 16S rRNA gene (*Schloss, 2010*), we realigned the FASTA sequences to a SILVA reference alignment to identify the V4 region of the sequences.

### Calculation of Matthew's Correlation Coefficient (MCC)

The MCC was calculated by two approaches in this study using only the dereplicated sequence lists. First, we calculated the MCC to determine the stability of OTU assignments following the approach of the He study. We assumed that the clusters obtained from the 30 randomized full datasets were correct. We counted the number of sequence pairs that were in the same OTU for the subsetted dataset and the full dataset (i.e., true positives; TP), that were in different OTUs for the subsetted dataset and the full dataset (i.e., true negatives; TN), that were in the same OTU for the subsetted dataset and different OTUs in the full dataset (i.e., false positives; FP), and that were in different OTUs for the subsetted dataset and the same OTU in the full dataset (i.e., false negatives; FN). For each set of 30 random subsamplings of the dataset, we counted these parameters against the 30 randomizations of the full dataset. This gave 900 comparisons for each fraction of sequences being used in the analysis. The Matthew's correlation coefficient was then calculated as:

$$\text{MCC} = \frac{\text{TP} \times \text{TN} - \text{FP} \times \text{FN}}{\sqrt{(\text{TP} + \text{FP})(\text{TP} + \text{FN})(\text{TN} + \text{FP})(\text{TN} + \text{FN})}}.$$

Second, we calculated the MCC to determine the quality of the clusterings as previously described (*Schloss & Westcott, 2011*). Briefly, we compared the OTU assignments for pairs of sequences to the distance matrix that was calculated between all pairs of aligned sequences. For each dataset that was clustered, those pairs of sequences that were in the same OTU and had a distance less than 3% were TPs, those that were in different OTUs and had a distance greater than 3% were TNs, those that were in the same OTU and had a distance greater than 3% were FPs, and those that were in different OTUs and had a distance less than 3% were FNs. The MCC was counted for each dataset using the formula above as implemented in the `sens.spec` command in mothur. To judge the quality of the Swarm-generated OTU assignments we calculated the MCC value using thresholds incremented by 1% between 0 and 5% and selected the threshold that provided the optimal MCC value.

### Software availability

A reproducible workflow including all scripts and this manuscript as a literate program­ming document are available at https://github.com/SchlossLab/Schloss_Cluster_PeerJ_2015. The workflow utilized QIIME (v.1.9.1; *Caporaso et al., 2010*), mothur (v.1.37.0; *Schloss et al., 2009*), USEARCH (v.6.1; *Edgar, 2010*), VSEARCH (v.1.5.0; *Rognes et al., 2015*), Swarm (v.2.1.1; *Mahé et al., 2014*), and R (v.3.2.0; *R Core Team, 2015*). The SL, AL, and CL methods are called nearest neighbor (NN), average neighbor (AN), and furthest

neighbor (FN) in mothur; we have used the terminology from the He study to minimize confusion. The knitr (v.1.10.5; *Xie, 2013*), Rcpp (v. 0.11.6; *Eddelbuettel, 2013*), rentrez (v. 1.0.0; *Winter, Chamberlain & Guangchun, 2015*), and jsonlite (v. 0.9.16; *Ooms, 2014*) packages were used within R.

# RESULTS AND DISCUSSION

## Summary and replication of He study

We obtained the Canadian soil dataset from *Roesch et al. (2007)* and processed the sequences as described by He and colleagues. Using these data, we reconsidered three of the more critical analyses performed in the He study.

First, we sought to quantify whether the OTU assignments observed for a subset of the data represented the same assignments that were found with the full dataset. The He study used the MCC to quantify the degree to which pairs of sequences were in the same OTUs in subsampled and full datasets. A more robust approach would utilize metrics that quantify the mutual information held between two sets of clusterings and has been applied to assess inter-method variation in OTU composition (*Schmidt, Rodrigues & Mering, 2014b*). To maintain consistency with the original He study, we also calculated the MCC value as they described. The He study found that when they used the open and closed-reference methods the OTUs formed using the subsetted data most closely resembled those of the full dataset. Among the *de novo* methods they observed that the AGC method generated the most stable OTUs followed by the single linkage (SL), DGC, complete linkage (CL), and average linkage (AL) methods. We first calculated the MCC for the OTU assignments generated by each of the clustering methods using 20, 40, 60, and 80% of the sequences relative to the OTU composition formed by the methods using the full dataset (see 'Methods' for description; Fig. 1A). Similar to the He study, we replicated each method and subsampled to the desired fraction of the dataset 30 times. Multiple subsamplings were necessary because a random number generator is used in some of the methods to break ties where pairs of sequences have the same distance between them. Across these sequencing depths, we observed that the stability of the OTUs generated by the SL and CL methods were highly sensitive to sampling effort relative to the OTUs generated by the AL, AGC, and DGC methods (Fig. 1A). Our results (Fig. 1B) largely confirmed those of Fig. 4C in the He study with one notable exception. The He study observed a broad range of MCC values among their AL replicates when analyzing OTUs generated using 60% of the data. This result appeared out of character and was not explained by the authors. They observed a mean MCC value of approximately 0.63 (95% CI [0.15–0.75]). In contrast, we observed a mean value of 0.93 (95% CI [0.91–0.95]). This result indicates that the AL assignments were far more stable than indicated in the He study. Regardless, although the assignments are quite stable, it does support the assertion that the OTU assignments observed for the subset of the data do not perfectly match the assignments that were found with the full dataset as they did with the reference-based methods; however, the significance of these differences is unclear.

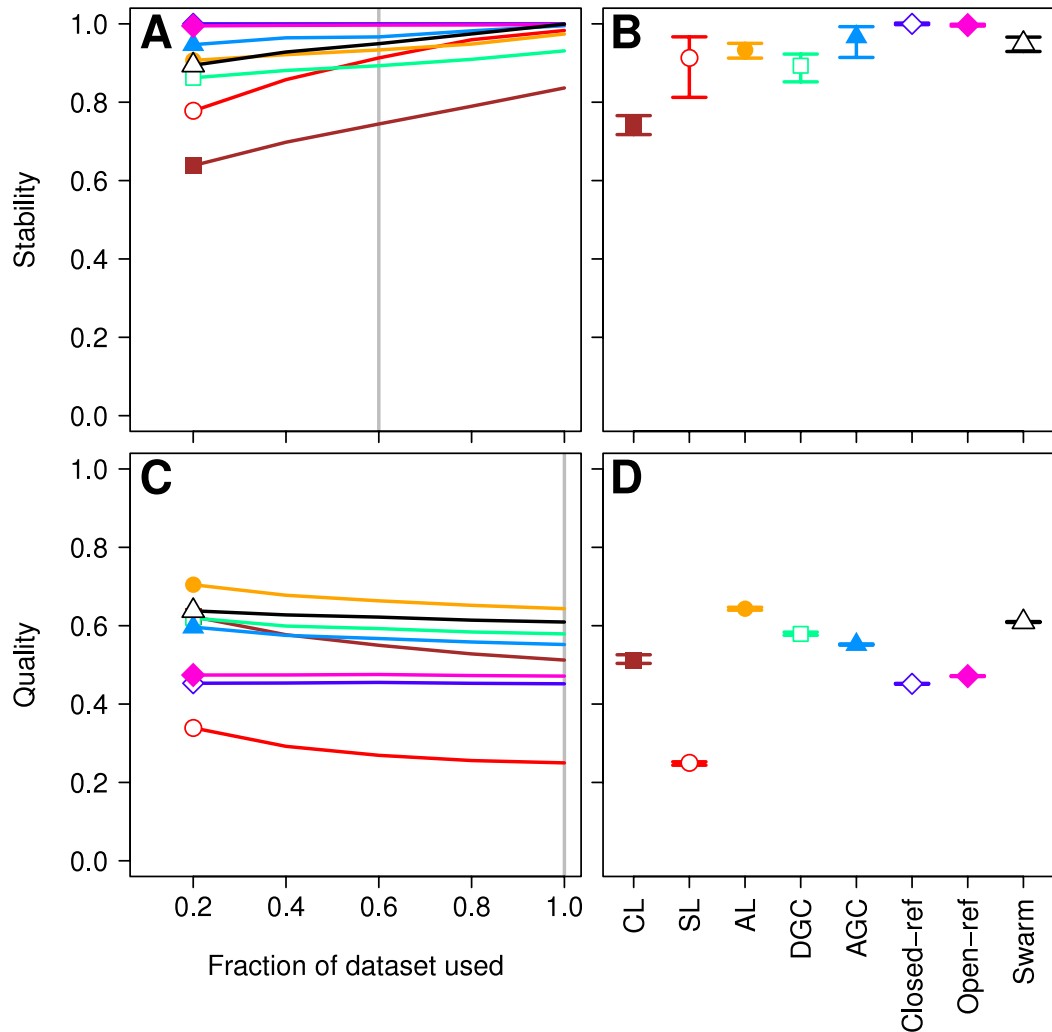

**Figure 1 Comparison of the stability (A, B) and quality (C, D) of *de novo* and reference-based clustering methods using the Canadian soil dataset.** The average stability of the OTUs was determined by calculating the MCC with respect to the OTU assignments for the full dataset using varying sized subsamples. The quality of the OTUs was determined by calculating the MCC with respect to the distances between the sequences using varying sized subsamples. Thirty randomizations were performed for each fraction of the dataset and the average and 95% confidence interval are presented when using 60% of the data. The vertical gray lines in A and C indicates the fraction of the dataset represented in B and D, respectively. The color and shape of the plotting symbol is the same between the different panels and is described along the *x*-axis of panel D. The optimum threshold for the Swarm-generated assignments was 3%.

Second, the He study and the original Roesch study showed that rarefaction curves calculated using CL-generated OTU assignments obtained using a subsample of the sequencing data did not overlap with rarefaction curves generated using OTU assignments generated from the full dataset. The He and Roesch studies both found that the CL method produced fewer OTUs in the subset than in the rarefied data. In addition, the He study found that the SL method produced more OTUs, the AGC produced fewer, and the

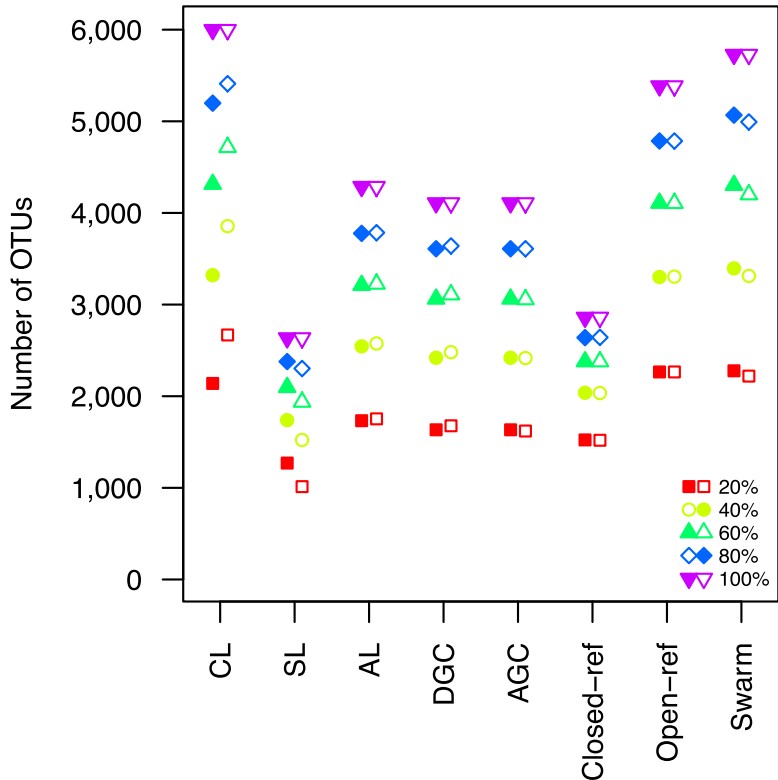

**Figure 2 The clustering methods varied in their ability to generate the same number of OTUs using a subset of the data as were observed when the full dataset was rarefied.** The subsetted data are depicted by closed circles and the data from the rarefied full dataset is depicted by the open circles.

other methods produced similar numbers of OTUs than expected when comparing the subsetted data to the rarefied data. Our results support those of these previous studies (Fig. 2). It was clear that inter-method differences were generally more pronounced than the differences observed between rarefying from the full dataset and from clustering the subsetted data. The number of OTUs observed was largest using the CL method, followed by the open-reference method. The AL, AGC, and DGC methods all provided comparable numbers of OTUs. Finally, the closed-reference and SL methods generated the fewest OTUs.

Third, the authors attempted to describe the effects of the OTU assignment instability on comparisons of communities. They used Adonis to test whether the community structure represented in subsetted communities resembled that of the full dataset when only using the unstable OTUs (*Anderson, 2001*). Although they were able to detect significant *p*-values, they appeared to be of marginal biological significance. Adonis R statistics close to zero indicate the community structures from the full and subsetted datasets overlapped while values of one indicate the communities are completely different. The He study observed adonis R statistics of 0.02 (closed-reference), 0.03 (open-reference), 0.07 (CL, AGC, DGC), and 0.16 (SL and AL). Regardless of the statistical or biological significance of these results, the analysis was tautological since, by definition, representing

communities based on their unstable OTUs would yield differences. Furthermore, the *de novo* and open-reference approaches do not consistently label the OTUs that sequences belong to when the clustering methods are run multiple times with different random number seeds. To overcome this, the authors selected representative sequences from each OTU and used those representative sequences to link OTU assignments between the different sized sequence sets. It was not surprising that the only analysis that did not provide a significant *p*-value was for the closed-reference analysis, which is the only analysis that provides consistent OTU labels. Finally, the authors built off of this analysis to count the number of OTUs that were differentially represented between the subsetted and full datasets by each method. This entire analysis assumed that the OTUs generated using the full dataset were correct, which was an unsubstantiated assumption since the authors did not assess the quality of the OTU assignments.

This re-analysis of the He study raised five complementary questions. First, how do the various methods vary in the quality of their OTU assignments? Second, how generalizable are these results to modern datasets generated using a large number of sequences that were deeply sequenced? Third, how does the stability and quality of OTU assignments generated by new methods compare to those analyzed in the He study? Fourth, are there open-source alternatives to USEARCH that perform just as well? Finally, although the stability of reference-based methods did not appear to be impacted by the input order of the sequences to be assigned to OTUs, is the stability of reference-based methods impacted by the order of the reference sequences? In the remainder of the 'Results and Discussion' we address each of these questions.

## How do the various methods vary in the quality of their OTU assignments?

More important than the stability of OTUs is whether sequences are assigned to the correct OTUs. A method can generate highly stable OTUs, but the OTU assignments may be meaningless if they poorly represent the specified cutoff and the actual distance between the sequences. To assess the quality of OTU assignments by the various methods, we made use of the pairwise distance between the unique sequences to count the number of true positives and negatives and the number of false positives and negatives for each method and sampling depth. Counting the frequency of these different classes allowed us to judge how each method balanced the ratio of true positives and negatives to false positives and negatives using the MCC. We used the average MCC value as a measure of a method's quality and its variation as a measure of its consistency. We made three important observations. First, each of the *de novo* methods varied in how sensitive their MCC values were to additional sequences (Fig. 1C). The SL and CL methods were the most sensitive; however, the quality of the OTU assignments did not meaningfully differ when 80 or 100% of the data were assigned to OTUs using the *de novo* methods. Second, the AL method had higher MCC values than the other methods followed by DGC, AGC, CL, open-reference, and closed-reference, and SL (Fig. 1D). Third, with the possible exception of the CL method, the MCC values for each of the methods only demonstrated a small

amount of variation between runs of the method with a different ordering of the input sequences. This indicates that although there may be variation between executions of the same method, they produced OTU assignments that were of equal quality. Revisiting the concept of stability, we question the value of obtaining stable OTUs when the full dataset is not optimally assigned to OTUs. Our analysis indicates that the most optimal method for assigning the Canadian soils sequences to OTUs using a 97% threshold was the AL method.

## How generalizable are these results to modern datasets generated using a large number of sequences that were deeply sequenced?

Three factors make the Canadian soil dataset less than desirable to evaluate clustering methods. First, it was one of the earliest 16S rRNA gene sequence datasets published using the 454 FLX platform. Developments in sequencing technology now permit the sequencing of millions of sequences for a study. In addition, because the original Phred quality scores and flowgram data are not available, it was not possible for us to adequately remove sequencing errors (*Schloss, Gevers & Westcott, 2011*). The large number of sequencing errors that one would expect to remain in the dataset are likely to negatively affect the performance of all of the clustering methods. Second, the dataset used in the He study covered the V9 region of the 16S rRNA gene. For a variety of reasons, this region is not well represented in databases, including the reference database used by the closed and open-reference methods. Of the 99,322 sequences in the default QIIME database, only 48,824 fully cover the V9 region. In contrast, 99,310 of the sequences fully covered the V4 region. Inadequate coverage of the V9 region would adversely affect the ability of the reference-based methods to assign sequences to OTUs. Third, our previous analysis has shown that the V9 region evolves at a rate much slower than the rest of the gene (*Schloss, 2010*). With these points in mind, we compared the clustering assignment for each of these methods using a time series experiment that was obtained using mouse feces (*Schloss et al., 2012*; *Kozich et al., 2013*). The MiSeq platform was used to generate 2,825,000 sequences from the V4 region of the 16S rRNA gene of 360 samples. Parallel sequencing of a mock community indicated that the sequencing error rate was approximately 0.02% (*Kozich et al., 2013*). Although no dataset is perfect for exhaustively testing these clustering methods, this dataset was useful for demonstrating several points. First, when using 60% of the data, the stability relationships amongst the different methods were similar to what we observed using the Canadian soil dataset (Figs. 3A and 3B). With the exception of the clusters generated using CL, the methods all performed very well with stabilities greater than 0.91. Second, the MCC values calculated relative to the distances between sequences were generally higher than was observed for the Canadian soil dataset for all of the methods except the CL and SL methods. Surprisingly, the MCC values for the DGC (0.77) and AGC (0.76) methods were comparable to the AL method (0.76; Figs. 3C and 3D). This result suggests that the optimal method is likely to be database-dependent.

Finally, as was observed with the Canadian soil dataset, there was little variation in the MCC values observed among the 30 randomizations. Therefore, although the methods have a stochastic component, the OTU assignments did not vary meaningfully between

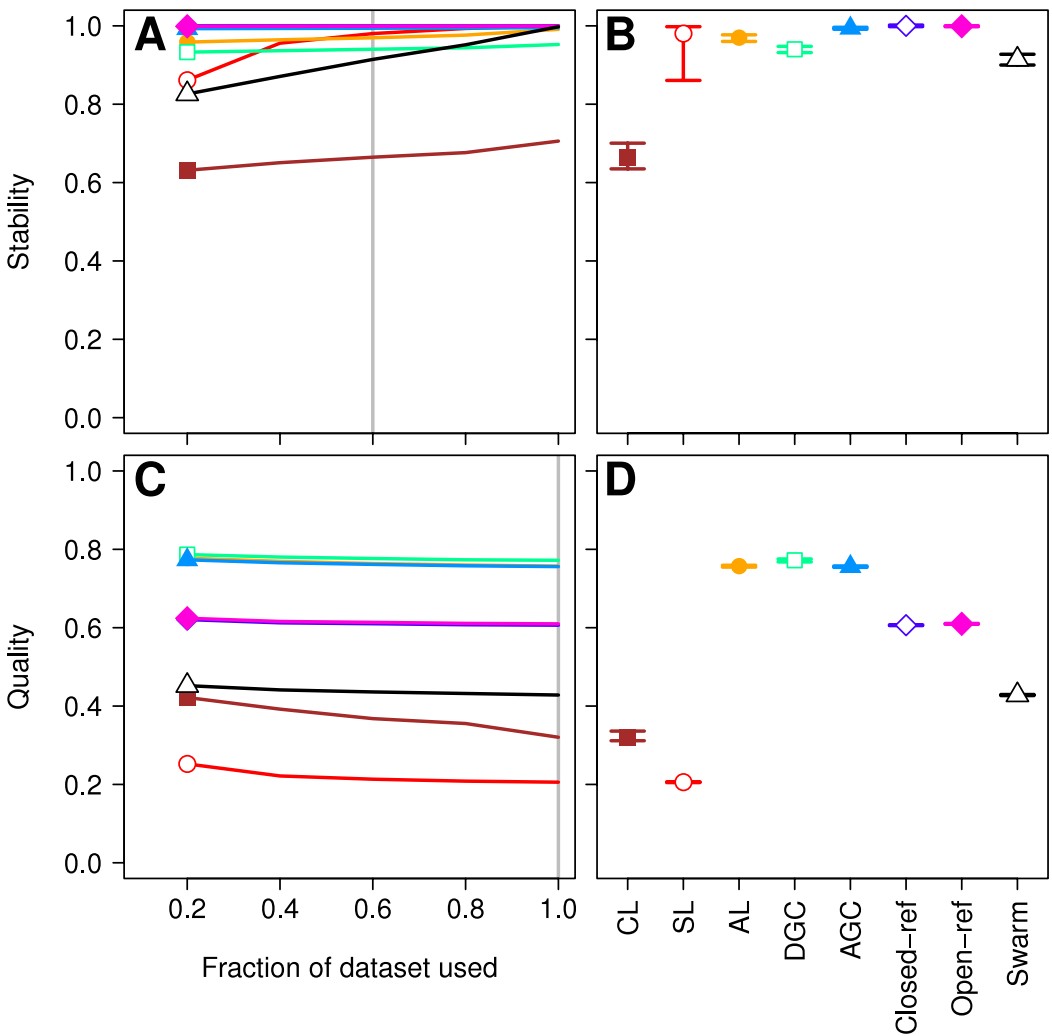

**Figure 3 Comparison of the stability (A, B) and quality (C, D) of *de novo* and reference-based clustering methods using the murine dataset.** The average stability of the OTUs was determined by calculating the MCC with respect to the OTU assignments for the full dataset using varying sized subsamples. The quality of the OTUs was determined by calculating the MCC with respect to the distances between the sequences using varying sized subsamples. Thirty randomizations were performed for each fraction of the dataset and the average and 95% confidence interval are presented when using 60% of the data. The vertical gray lines in A and C indicates the fraction of the dataset represented in B and D, respectively. The color and shape of the plotting symbol is the same between the different panels and is described along the *x*-axis of panel D. The optimum threshold for the Swarm-generated assignments was 2%.

runs. The results from both the Canadian soil and murine microbiota datasets demonstrate that the *de novo* methods can generate stable OTU assignments and that the overall quality of the assignments were consistent. Most importantly, these analyses demonstrate that the OTU assignments using the AL, AGC, and DGC *de novo* methods were consistently better than either of the reference-based methods.

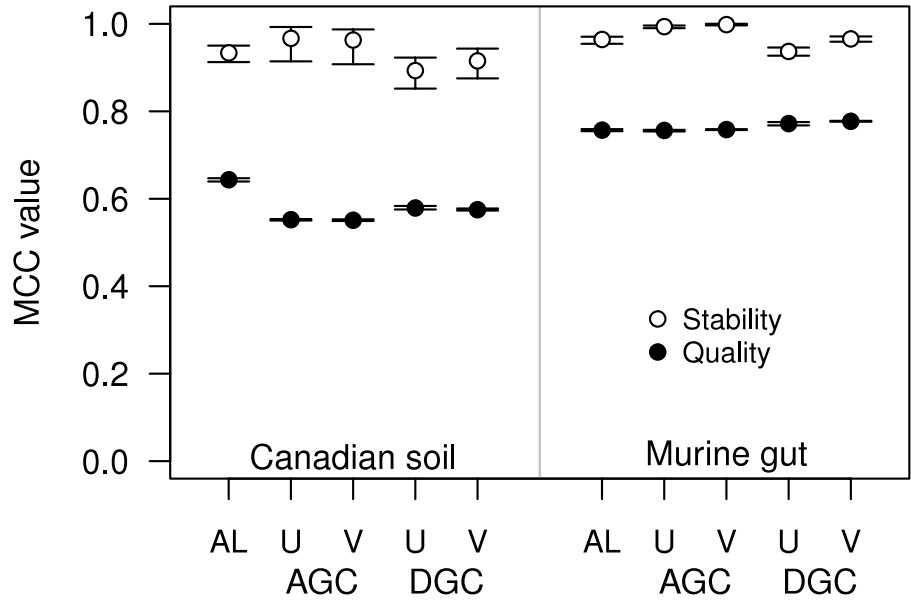

**Figure 4** **The stability and quality of USEARCH and VSEARCH OTUs generated by the AGC and DGC methods were similar.** The stability of the OTUs was determined by calculating the MCC for OTUs calculated using 60% of the data relative to the OTU assignments for the full dataset. The quality of the OTUs was determined by calculating the MCC of the OTUs calculated using the full dataset with respect to the distances between the sequences. The error bars represent the 95% confidence interval across the 30 randomizations.

## How does the stability and quality of OTU assignments generated by new methods compare to those analyzed in the He study?

The Swarm algorithm is a recently proposed *de novo* method for assigning sequences to OTUs that uses user-defined parameters to break up chains generated by SL clustering (*Mahé et al., 2014*). Swarm was originally validated by comparing the results against the expected clusters formed based on the taxonomic composition of a mock community. Similar to the authors of the He study, the Swarm developers suggest that methods are needed that are insensitive to input order. Use of Swarm on the Canadian soil and murine datasets demonstrated that similar to the other *de novo* methods, Swarm's OTU assignments changed as sequences were added (Figs. 1A and 3A). When we compared the OTU assignments for both datasets when using all of the sequence data, the variation in MCC values across the 30 randomizations were not meaningfully different (Figs. 1D and 3D). Most importantly, when we selected the distance threshold that optimized the MCC value, the quality of the OTU assignments was close to that of the AL assignments when using the Canadian soil dataset and considerably worse than that of the murine dataset (Figs. 1D and 3D). Interestingly, the distance thresholds that resulted in the largest MCC values were 3 and 2% for the Canadian soil and murine datasets, respectively. This suggests that distance-based OTU definitions are not consistent across datasets when using the Swarm algorithm, although they do appear to be within the neighborhood of 3%. Finally, the Swarm developers indicated that hierarchical *de novo* algorithms were too impractical to use on large MiSeq-generated datasets. Our ability to apply AL to the large

mouse dataset and even larger datasets suggests that it is not necessary to sacrifice OTU assignment quality for speed (e.g., *Schubert, Sinani & Schloss, 2015*; *Zackular et al., 2015*).

## Are there open-source alternatives to USEARCH that perform just as well?

For some datasets the AGC and DGC methods appear to perform as well or better than the hierarchical clustering methods. As originally described in the He study, the AGC and DGC methods utilized the USEARCH program and the DGC method is used for clustering in UPARSE (*Edgar, 2010*; *Edgar, 2013*). The source code for USEARCH is not publicly available and only the 32-bit executables are available for free to academic users. Access for non-academic users and those needing the 64-bit version is available commercially from the developer. An alternative to USEARCH is VSEARCH, which is being developed in parallel to USEARCH as an open-source alternative. One subtle difference between the two programs is that USEARCH employs a heuristic to generate candidate alignments whereas VSEARCH generates the actual global alignments. The VSEARCH developers claim that this difference enhances the sensitivity of VSEARCH relative to USEARCH. Using the two datasets, we determined whether the AGC and DGC methods, as implemented by the two programs, yielded OTU assignments of similar quality. In general the overall trends that we observed with the USEARCH-version of AGC and DGC were also observed with the VSEARCH-version of the methods (Fig. 4). When we compared the two implementations of the AGC and DGC methods, the OTUs generated by the VSEARCH-version of the methods were as stable or more stable than the USEARCH-version when using 60% of the datasets. In addition, the MCC values for the entire datasets, calculated relative to the distance matrix, were virtually indistinguishable. These results are a strong indication that VSEARCH is a suitable and possibly better option for executing the AGC and DGC methods.

## Is the stability of reference-based methods impacted by the order of the reference sequences?

The He study and our replication attempt validated that the closed-reference method generated perfectly stable OTUs. This was unsurprising since, by definition, the method is designed to return one-to-one mapping of reads to a reference. Furthermore, because it treats the input sequences independently the input order or use of a random number generator is not an issue. An important test that was not performed in the He study was to determine whether the clustering was sensitive to the order of the sequences in the database. The default database used in QIIME, which was also used in the He study, contains full-length sequences that are at most 97% similar to each other. We randomized the order of the reference sequences 30 times and used them to carry out the closed-reference method with the full murine dataset, which contained 32,106 unique sequences (Fig. 5). Surprisingly, we observed that the number of OTUs generated was not the same in each of the randomizations. On average there were 28,059 sequences that mapped to a reference OTU per randomization (range from 28,007 to 28,111). The original ordering of the reference resulted in 27,876 sequences being mapped, less than the

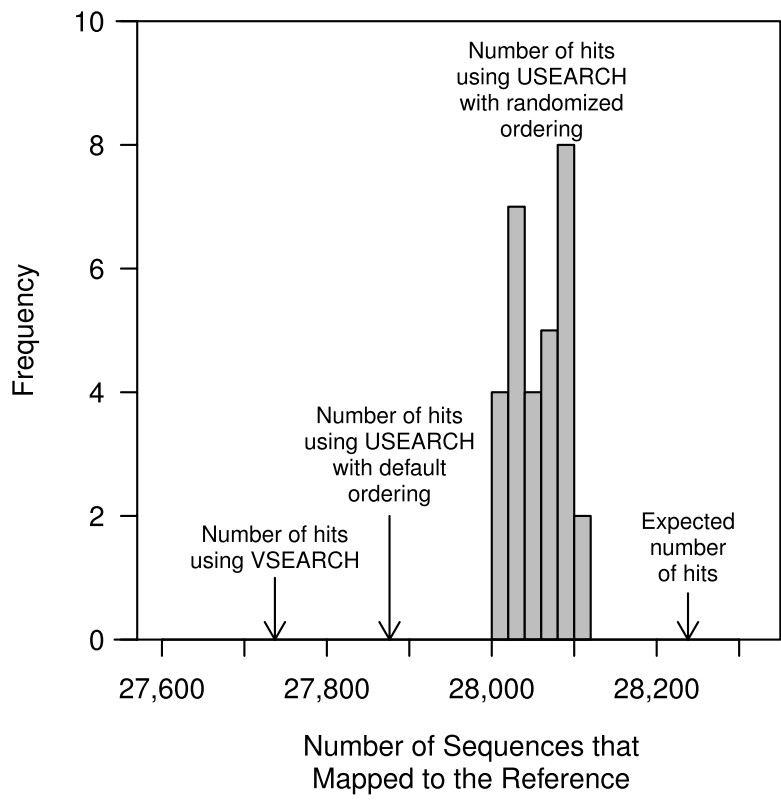

**Figure 5 The number of closed-reference OTUs observed in the murine dataset when using USEARCH, VSEARCH, and without a heuristic.** In addition to the default ordering of the references provided with the QIIME package, the reference sequences were randomized 30 times; the order of the murine dataset was not randomized. Regardless of whether the default or randomized ordering was used, the number of OTUs generated using VSEARCH did not differ. The non-heuristic approach calculated the exact distance between the murine sequences and the reference sequences and assigned the sequences to the reference with the smallest distance.

minimum observed number of mapped sequences when the references were randomized. This surprising result was likely due to the performance of the USEARCH heuristic. To test this further, we substituted VSEARCH for USEARCH in the closed-reference method. When we used VSEARCH the original ordering of the reference sequences and all randomizations were able to map 27,737 sequences to reference OTUs. When we calculated the true distance between each of the murine sequences and the references, we were able to map 28,238 of the murine sequences to the reference sequences when using a 97% similarity threshold without the use of a heuristic. This result indicates that the closed reference approach, whether using USEARCH or VSEARCH, does not exhaustively or accurately map reads to the closest reference. To quantify this further, we calculated the MCC for the USEARCH and VSEARCH assignments relative to the assignments using the non-heuristic approach. Using USEARCH the average MCC was 0.78 (range: 0.75–0.80) and using VSEARCH the average MCC was 0.65 (range: 0.64–0.66). The two methods had similar sensitivities (USEARCH: 0.98 and VSEARCH: 0.97), but the USEARCH specificity (0.73) was considerably higher than VSEARCH (0.60). Overall, these results indicate that

although heuristic approaches may be fast, they do a poor job of mapping reads to the correct reference sequence relative to non-heuristic approaches.

We also observed that regardless of whether we used USEARCH or VSEARCH, the reference OTU labels that were assigned to each OTU differed between randomizations. When we used USEARCH to perform closed-reference clustering, an average of 57.38% of the labels were shared between pairs of the 30 randomizations (range = 56.14–59.55%). If we instead used VSEARCH an average of 56.23% of the labels were shared between pairs of the 30 randomizations (range = 53.48–59.12%). To better understand this result, we further analyzed QIIME's reference database. We hypothesized that within a given region there would be sequences that were more than 97% similar and possibly identical to each other. When a sequence was used to search the randomized databases, it would encounter a different reference sequence as the first match with each randomization. Among the 99,310 reference sequences that fully overlap the V4 region, there were 7,785 pairs of sequences that were more than 97% similar to each other over the full length of the 16S rRNA gene. When the extracted V4 sequences were dereplicated, we identified 88,347 unique sequences. Among these dereplicated V4 sequences there were 311,430 pairs of sequences that were more than 97% similar to each other. The presence of duplicate and highly similar V4 reference sequences explains the lack of labeling stability when using either USEARCH or VSEARCH to carry out the closed-reference method. We suspect that the reference database was designed to only include sequences that were at most 97% similar to each other as a way to overcome the limitations of the USEARCH search heuristic.

Beyond comparing the abundance of specific OTUs across samples, the reference database is used in the open and closed-reference methods to generate OTU labels that can be used in several downstream applications. It is commonly used to extract information from a reference phylogenetic tree to carrying out UniFrac-based analyses (*Hamady, Lozupone & Knight, 2009*) and to identify reference genomes for performing analyses such as PICRUSt (*Langille et al., 2013*). Because these downstream applications depend on the correct and unique labeling of the OTUs, the lack of label stability is problematic. As one illustration of the effects that incorrect labels would have on an analysis, we asked whether the duplicate sequences had the same taxonomies. Among the 3,132 V4 reference sequences that had one duplicate, 443 had discordant taxonomies. Furthermore, among those 1,699 V4 reference sequences with two or more duplicates, 698 had discordant taxonomies. Two V4 reference sequences mapped to 30 and 10 duplicate sequences and both contained 7 different taxonomies. Among the V4 sequences within the database, there was also a sequence that had 131 duplicates and represented 5 different taxonomies. When we analyzed the 28,238 sequences that mapped to the V4 reference sequences using a non-heuristic approach, we observed that 18,315 of the sequences mapped to more than one reference sequence. Of these sequences, 13,378 (73.04%) mapped to references that were identical over the V4 region and 4,937 (26.96%) mapped equally well to two or more references that were not identical over the V4 region. Among the combined 18,315 sequences that mapped to multiple reference sequences, the taxonomy of the multiple

reference sequences conflicted for 3,637 (19.86%). Together, these results demonstrate some of the considerable problems with the reference-based clustering of sequences.

## CONCLUSIONS

It is worth noting that the analysis from the Roesch study that motivated the He study is not typical of microbial ecology studies. First, their analysis was based on a single soil sample. Researchers generally have dozens or hundreds of samples that are pooled and clustered together to enable comparison across samples. Second, all of the sequence data from these datasets is usually pooled for a single analysis. Rarely would a researcher rarefy their data prior to clustering since it can be more efficiently done after all of the data are assigned to OTUs. Third, the CL method used in the original Roesch study has since been shown to not generate optimal OTUs (*Schloss & Westcott, 2011*). As for the approach used in the He study, the value of identifying stable OTUs is unclear. Although there is concern that running the methods multiple times yields different clusterings, we have shown that there is little variation in their quality. This suggests that the different clusterings by the same method are equally good. Greater emphasis should be placed on obtaining an optimal balance between splitting similar sequences into separate OTUs and merging disparate sequences into the same OTU.

The approach of the current study quantified the effects of merging and splitting OTUs by using an objective metric. Through the use of the pairwise distances between sequences, we were able to use the MCC to demonstrate that, in general, the AL method was consistently the optimal method for each dataset, but that Swarm, AGC, and DGC sometimes perform as well as AL. At least for the murine dataset, Swarm also could be among the methods that performed poorly. It is impossible to obtain a clustering with no false positives or false negatives and the optimal method may vary by dataset. With this in mind, researchers are encouraged to calculate and report their MCC values and to use these values to justify using methods other than the AL. As an alternative to the He study's method of measuring stability, we propose using the variation in the quality of the clustering of the full dataset. Given the tight 95% confidence intervals shown in Figs. 1D and 3D, with the exception of CL, it is clear that this variation is quite small. This indicates that although the order of the sequences being clustered can affect the actual cluster assignments, the quality of those different clusterings is not meaningfully different.

Our analysis of those methods that implemented USEARCH as a method for clustering sequences revealed that its heuristic limited its specificity. When we replaced USEARCH with VSEARCH, the *de novo* clustering quality was as good or better. Although there may be parameters in USEARCH that can be tuned to improve the heuristic, these parameters are likely dataset dependent. Based on the data presented in this study, its availability as an open source, and free program, VSEARCH should replace USEARCH in the *de novo* clustering methods; however, USEARCH performed better than VSEARCH for closed-reference clustering. Furthermore, although not tested in our study, VSEARCH can be parallelized leading to potentially significant improvements in speed. Although USEARCH and VSEARCH do not utilize aligned sequences, it is important to note that

a sequence curation pipeline including denoising, alignment, trimming to a consistent region of the 16S rRNA gene, and chimera checking are critical to making proper inferences (*Schloss, Gevers & Westcott, 2011*; *Schloss, 2012*; *Kozich et al., 2013*).

We have assessed the ability of reference-based clustering methods to capture the actual distance between the sequences in a dataset in parallel with *de novo* methods. Several studies have lauded both the open and closed-reference approaches for generating reproducible clusterings (*Navas-Molina et al., 2013*; *Rideout et al., 2014*; *He et al., 2015*), yet we have shown that both reference-based approaches did a poor job of representing the distance between the sequences compared to the *de novo* approaches. Although the OTU assignments are reproducible and stable across a range of library sizes, the reference-based OTU assignments are a poor representation of the data. We also observed that the assignments were not actually reproducible when the order of the reference sequences was randomized. When USEARCH was used, the actual number of sequences that mapped to the reference changed depended on the order of the reference. Perhaps most alarming was that the default order of the database provided the worst MCC of any of the randomizations we attempted. This has the potential to introduce a systematic bias rather than a random error. Even when we used VSEARCH to perform closed-reference clustering and were able to obtain consistent clusterings, we observed that the labels on the OTUs differed between randomizations. Because the OTU labels are frequently used to identify representative sequences for those OTUs, variation in labels, often representing different taxonomic groups, will have a detrimental effect on the interpretation of downstream analyses.

Because the open-reference method is a hybrid of the closed-reference and DGC methods, it is also negatively affected by the various problems using USEARCH. An added problem with the open-reference method is that the two phases of the method employ different thresholds to define its OTUs. In the closed-reference step, sequences must be within a threshold of a reference to be in the same OTU. This means that in the worst case scenario two sequences that are 97% similar to a reference, and are joined into the same OTU, may only be 94% similar to each other. In the DGC step, the goal is to approximate the AL method which requires that, on average, the sequences within an OTU are, on average, 97% similar to each other. The end result of the open-reference approach is that sequences that are similar to previously observed sequences are clustered with one threshold while those that are not similar to previously observed sequences are clustered with a different threshold.

As the throughput of sequencing technologies have improved, development of clustering algorithms must continue to keep pace. *De novo* clustering methods are considerably slower and more computationally intensive than reference-based methods and the greedy *de novo* methods are faster than the hierarchical methods. In our experience (*Kozich et al., 2013*), the most significant detriment to execution speed of the *de novo* methods has been the inadequate removal of sequencing error and chimeras. As the rate of sequencing error increases so do the number of unique sequences that must be clustered. The speed of the *de novo* methods scales approximately quadratically, so that doubling the number of sequences results in a four-fold increase in the time required to

execute the method. The rapid expansion in sequencing throughput has been likened to the Red Queen in Lewis Carroll's *Through the Looking-Glass* who must run in place to keep up with her changing surroundings (*Schloss et al., 2009*). Microbial ecologists must continue to refine clustering methods to better handle the size of their growing datasets, but they must also take steps to improve the quality of the underlying data. Ultimately, objective standards must be applied to assess the quality of the data and the quality of OTU clustering.

### Funding
This study was supported by grants from the NIH (R01GM099514 and P30DK034933). The funders had no role in study design, data collection and analysis, decision to publish, or preparation of the manuscript.

### Grant Disclosures
The following grant information was disclosed by the authors:
NIH: R01GM099514, P30DK034933.

### Competing Interests
The authors declare there are no competing interests.

### Author Contributions
- Sarah L. Westcott conceived and designed the experiments, performed the experiments, analyzed the data, contributed reagents/materials/analysis tools, reviewed drafts of the paper.
- Patrick D. Schloss conceived and designed the experiments, performed the experiments, analyzed the data, contributed reagents/materials/analysis tools, wrote the paper, prepared figures and/or tables, reviewed drafts of the paper.

### Data Availability
The complete data analysis pipeline is available at https://github.com/SchlossLab/Schloss_Cluster_PeerJ_2015.

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
