# Peer review of "De novo clustering methods outperform reference-based methods for assigning 16S rRNA gene sequences to operational taxonomic units"

_PeerJ, doi:10.7717/peerj.1487_

## Round 0.1 · original submission · Major Revisions

Dear authors,

Thank you very much for your effort to make your analyses easy to reproduce, and follow!

I agree with the salient point that is made by both reviewers: this is a well-written and timely work with many useful insights. Their comments and concerns are attached to this email.

In your revision, please pay extra attention to the Reviewer 1's criticism on the diverse terminology used throughout the manuscript (i.e, stability, rigor, robustness, quality, accuracy). The Reviewer 2 makes noteworthy suggestions on how to assess the quality of OTUs (MCC vs. AMI/NMI/ARI), and suggests additional citations for a more complete representation of the previous work in the field. Also, please consider the language issues raised by both reviewers, and remove harsh statements from your revised manuscript.

Thank you very much again,

Reviewer 1 ·

Basic reporting

Code to reproduce the results has been provided in a github repository.

See general comments.

Experimental design

See general comments.

Validity of the findings

See general comments.

Additional comments

This paper evaluates the "stability" and "quality" of several different OTU clustering methods. "Stability" is defined here as the consistency between OTU assignments made on subsets of the data and the full dataset. "Quality" is defined here as assigning sequencing reads that are <3% different to the same OTU, and sequencing reads >3% different to different OTUs. The authors broad conclusions are that while reference based clustering is more "stable", that hierarchical de novo methods, particularly average linkage, produce OTUs of higher "quality". The authors also make an important observation that the OTU assignments of commonly used reference methods depend on the ordering of the reference database.

This paper is largely a response to He et al. "Stability of operational taxonomic units: an important but neglected property for analyzing microbial diversity" which argued for reference based methods based on the "stability" of their OTU assignments.
* * *
This paper is well-written, makes several useful observations, and is clearly suitable for publication. It can be improved by certain clarifications. I recommend its acceptance with minor revisions, as detailed below.

More important points are preceded by an asterisk.
* * *
*Terminology: Over the course of the paper, (at least) five different terms are used to refer to the goodness of a clustering: stability, rigor, robustness, quality and accuracy. Yet only two distinct concepts are explored (stability and quality/accuracy). I strongly recommend that the words "rigorous" and "robust" be struck from the manuscript entirely. They are never defined, are unclear, and the first impression they give does not correspond well to the actual results they are being used to describe.

Abstract...

L5: replace "rigorous" with something else, eg. "optimal", or even "best". Rigor is not being evaluated here.

L7-L8: Recommend introducing "quality" here alongside its initial operational definition.

L16: replace "robust". Robust is a particularly poor choice of word here, as it brings to mind stability, which is not what is being referred to.

*L20-L21: USEARCH -> USEARCH and VSEARCH. According to your results, both methods produced OTU assignments that depended on the ordering of the reference database (~55-60% label consistency between reorderings of the database).

L27: replace "rigorous". What is described in this paper is that the de novo methods produce OTU clusterings that coincide better with the pairwise distances between the sequencing reads.

Introduction...

L46-47: Move the last sentence of this paragraph to the first sentence of the next paragraph.

L57-59: This sentence does not follow from the previous (don't need "Therefore,") and is not necessarily a shortcoming, but is rather an observation. All methods other than complete-linkage can join sequences together that are are less than 97% similar.

L60: seuquence -> sequence. And drop "subtle".

*L64: The other major advantage of reference based methods is that the OTU assignments can be compared across experiments and even studies. This should be mentioned here.

L68: speed -> computational cost (or something similar)

L76: drop "however"

L80: Move last sentence ahead of previous sentence, as this follows from the reference independence property, not the ability to subsequently classify.

L85: Clarify that this is a potential issue with "current" or "common" open-reference methods. It is not an inherent issue with any open-reference approach, as one can use coherent OTU assignment strategies for both parts (eg. join reference sequence or open-de-novo cluster if within 3%).

*L95-105: Evaluation of accuracy on mock communities needs to be included in this discussion of previous methods to evaluate OTU methods.

L109: Those sequences that... -> Those pairs of sequence that... The unit of this analysis is pairs of sequences, not individual sequences.

L117: attempted to characterize -> characterized

L137-139: Either explain how the stability MCC is calculated, or reference the Methods at this point.

L141: Multiple subsamplings was -> were

L148-152: This is almost an extreme level of disagreement on this point between essentially identical AL analyses in the He study and this study. I find the He results rather unbelievable, but can the authors shed any further light on what might have happened? THe He study doesn't provide a full workflow, but did provide the command they executed: cluster(column=unique.dist,name=names,cutoff=0.05,precision=1000,method=average)

*L182-190: Comparing OTUs on the basis of their representative sequence is not "specious" and the He analysis was not "so poorly designed". The most abundant sequence in an OTU is a biologically meaningful object, and we would like to think that OTUs are biologically meaningful and not just abstract objects. This section needs to be removed, or rewritten with less polemic language and with an explanation of how comparing representative sequences can break down (for example for very low abundance OTUs).

L209: Remove "rigorous". You've used "accuracy" in this section (which appears identical to "quality" elsewhere, consider consolidating), so replace rigorous with accurate.

L243: the assignments are highly reproducible -> the overall quality of the assignments are consistent.
The reproducibility of the assignments themselves was not evaluated here.

L245: Remove "robust". The de novo methods produce clusterings that are more consistent with the pairwise separations of the individual sequencing reads. That does not correspond to any normal usage of the word robust.

L246: Evalution -> Evaluation

L246-249: I don't really understand this. Swarm implements single-linkage clustering, in a way that is very clever computationally, but that is still single-linkage clustering.

L251: method -> methods

L326-328: It would be useful to the reader to include the number of V4 covering reference sequences here.

L323-328: Numbers of sequences and numbers of pairs of sequences are mixed together here in a confusing fashion. It would be better to do one or the other (eg. 7,785 _pairs of sequences_ that were more than 97% similar... -> XXX sequences within 97% of another reference sequence...). This allows the reader to easily evaluate the fraction of the reference database this corresponds to.

L341-345: Is this referring to sequences duplicated over V4? Or over the whole 16S gene?

L354: "It is worth noting that the entire design of the He study was artificial." - This is unnecessarily polemic. There are ways in which the He study was not ideal, but essentially every OTU evaluation article uses an "artificial" design. The points being made here stand on their own, and are only obscured by this kind of language.

L362: replace "robust"

L364: replace "robust"

L390-395: It appears to me that the dependence on the order of the reference database has the potential to introduce _systematic_ rather than just random errors, which is somewhat troubling.

·

Basic reporting

In their manuscript, 'De novo clustering methods out-perform reference-based methods for assigning 16S rRNA gene sequences to operational taxonomic units', Schloss & Westcott report a series of quality benchmarks on several established and recent 16S sequence clustering approaches and methods. The authors critically reproduced and extended a recent study by He et al. (Microbiome, 2015); they provide arguments and data that OTU "stability" is not necessarily indicative of OTU "quality"; they provide OTU quality benchmarks for very recent algorithms/software (SWARM and VSEARCH); and they report a novel perspective and tests on reference-based OTU demarcation. I believe that this work has merit, that the science is generally sound (but see comments below) and that the manuscript is timely and important for the growing community of microbial ecologists.

My main concern is that the study design is somewhat disparate, as the individual reported findings are not inherently inter-connected (see also comments on "Validity of the Findings"). Also, while the data and results provided generally support the authors' conclusions, I am not convinced that benchmarks based on the Matthew's Correlation Coefficient alone are the most informative approach for all the tests conducted (again, see comments below). Finally, there are several issues regarding the general reporting in the manuscript that could make it hard to follow for non-expert readers.

Following the journal's recommendation, I have tried to organize all comments into the appropriate sections ("Basic Reporting", "Experimental Design" and "Validity of the Findings"). Within these sections, I tried to organize all major comments in decreasing order of importance and minor comments chronologically, following the manuscript. However, several points touch more than one of the specified comment sections, and I have tried to disentangle these as much as possible in order to achieve a readable review.


Comments on "Basic Reporting".

- The manuscript is overall very readable, clearly written, the global structure is clear and all individual sections are well structured. That being said, the structure of the study is also somewhat disparate: the individual parts stand somewhat independently from each other and do not form a fully coherent body of research. In the current form, the authors do not make it sufficiently clear how the first part of the manuscript (a critical reproduction of the study by He et al) relates to the second (quality benchmarks of the very recent methods SWARM and VSEARCH) and, in turn, to the third (general problems with reference-based OTU demarcation). While all of these findings are valid and interesting in themselves, the manuscript seems to be telling several unrelated stories at once. One general topic is a comparison of de novo vs reference-based OTUs, another is the question of OTU "stability" vs "accuracy" and a third thread refers to the clustering quality of novel methods, in particular a comparison of USEARCH to its open-source clone VSEARCH. As I said, the manuscript is very readable as is, but these findings are not sufficiently synthesized in the Conclusions, in my opinion.

- All Figures, in their current form, are really non-obvious to interpret. Frankly speaking, when I first read through the manuscript, I understood all results from the text but really struggled at interpreting the figures. I believe they would profit from a careful workover.
=> Figures 1&3 lack color legends (it took me some time to realize that colors in the left panel corresponded to the categorical axis in right panel). Also, being partially color-blind, I had a hard time distinguishing the thick lines in Fig 1&3, left panels. Ordinate axis labeling in Fig 1&3 was not intuitive on a first read-through; the meaning of "MCC relative to full dataset" and "to distances" was not clear, and only a close look at the corresponding Methods section (but not the Figure caption) helped resolve this.
=> In my opinion, Fig 2 presents the data in a non-intuitive way. I am not sure which point the authors are trying to emphasize here: the illustration draws the focus to the generally varying number in absolute OTU counts, but from the arguments in the text I believe that relative differences in counts between re-clustered sub-setted datasets and rarefied full datasets are more important. The latter, however, are not clear to interpret from the figure as is. Also, it is not clear to me why clustering methods are the categorical axis while down-sampling steps are scattered vertically. Moreover, a comparison of SWARM is missing from this figure for some reason (?). I would suggest to split this figure into two parts: part A providing more classical rarefaction curves per method (with sub-setted clustering points as overlay scatter) and part B providing the (log) relative OTU count between rarefaction and sub-setting across down-sampling steps.
=> Figure 4 lacks a caption that is explanatory beyond the figure title.

- The Introduction is generally very clear, concise and readable. However, there is a large body of previous work on testing methods for OTU clustering, and the authors focus on only a very limited subset of studies. While I think that a comprehensive review of all studies on OTU benchmarking ever published is certainly outside the scope of the present work, I believe that the manuscript would much profit from a more in-depth discussion of at least some. Other works possibly worth mentioning include, but are not limited to:
=> Huse et al., Env Microbiol, 2010, DOI: 10.1111/j.1462-2920.2010.02193.x
=> White et al., BMC Bioinformatics, 2010, doi:10.1186/1471-2105-11-152
=> Barriuso et al., BMC Bioinformatics, 2011, doi:10.1186/1471-2105-12-473, who indeed provided an early investigation into "output variability" of OTU clustering (Table 7)
=> Bonder et al., Bioinformatics, 2012, doi: 10.1093/bioinformatics/bts552
=> Koeppel & Wu, NAR, 2013, doi: 10.1093/nar/gkt241
=> Schmidt et al., PLOS Comp Biol, 2014, DOI: 10.1371/journal.pcbi.1003594
=> May et al., Bioinformatics, 2014, doi: 10.1093/bioinformatics/btu085
However, these are merely suggestions and the authors should not feel obliged to include any or all of these as new references in the Introduction – in particular, since I also pointed out one work of which I am co-author. Nevertheless, I do believe that the manuscript would generally profit from a discussion of (some) further previous work.

- The Introduction provides a detailed, accessible and balanced explanation of the three general approaches to OTU demarcation discussed in the paper: de novo, closed reference and open reference. Although the authors point to two comprehensive references which give further information on these points (Schloss & Westcott, 2011; Navas-Molina et al., 2013), I would suggest that the authors provide at least some visual/structural guidance to the reader (an overview figure or table). However, the manuscript is certainly understandable as is, so please consider this comment as a mere suggestion.

- Related to this, the terms "abundance-based" and "distance-based greedy clustering" (AGC and DGC) are not properly introduced. Although the authors point out that they stick with the terminology of the He et al paper, these terms need to be explained somewhere in the manuscript, preferably in the Introduction.

- Similarly, the last part in the Results and Discussion, "Problems with reference-based clustering [...]", could profit from an illustration. This part is very clearly written, and the presented results are striking (see also comments under "Validity of the Findings"). Nevertheless, a visual presentation of the data could additionally help to guide the reader through the conducted tests and illustrate just how striking the reported results are. Even a simple table summarizing the results would be helpful.

- The authors very clearly point out weaknesses in the criticized study by He et al; they support all their arguments with data and/or sound reasoning and do not, in my opinion, state any unfair or unsupported claims against the criticized work. Nevertheless, I personally felt that the wording is sometimes a bit harsh: e.g., p.9, l.190 ("this analysis was so poorly designed"); p.15, l.354 ("the entire design of the He study was artificial").


Minor comments:

- Please re-check for typos and grammar across the text (e.g., p.5, l.60 "seuquence"; p.5, l.69 "scale[s]"; p.6, l.88 "assigned"; p.7, l.137 "assess the calculated the MCC"; etc.)

- Several examples given throughout the text appear rather arbitrary. For example, p.5, l.59 ("may only be 94% similar", same in p.17, ll.402-3) or p.6, l.89 ("all sequences assigned to the Porphyromonadaceae would then be assigned to OTUs [...]") are out of context, the sudden mention of a concrete clustering threshold or an arbitrary bacterial family could confuse the reader more than illustrate the point being made.

- p.4, l.56. I believe that the provided reference for the statement that "the commonly used variable regions [...] do not evolve at the same rate" is not fully appropriate. Citing Schloss, Plos Comp Biol, 2010 here (as is done elsewhere in the text) seems more meaningful, as this study is arguably the most comprehensive to date regarding this point. Moreover, it is probably also worth mentioning the related contribution by Kim et al. (Journal Micobiol Methods, 2011, http://www.ncbi.nlm.nih.gov/pubmed/21047533).

- When enumerating the pros and cons of closed ref clustering (p.5, ll. 48f), several points in favor of this approach could be added: it is trivially parallelizable, has a low memory footprint and arguably gets better as reference databases improve (which is the case as more and more sequence data accumulates).

- p.6, l.93-4. This statement is phrased too strongly. The approach advocated by the authors (taxonomic classification, followed by de novo clustering within taxonomic bins) has been shown to be very useful in practice, but presenting it as an ideal compromise between closed reference and de novo approaches is an overstatement and potentially misleading. The authors, themselves, point out just a few lines later that taxonomic classification is often far from perfect (it could be added that it depends strongly on reference database quality and classifier performance); however, if the classification step is not perfect, the entire clustering outcome will be (strongly) affected. Others have suggested database-agnostic alternatives in a similar spirit, such as SL pre-clustering (Huse et al., Env Microbiol, 2010) or the kmer-pre-clustering implemented in ESPRIT-Tree (Cai & Sun, NAR, 2011) which would be worth mentioning here.

- p.6, ll.109ff. This paragraph explaining the MCC (in detail) feels a little out of place here; I suggest to relocate it to the (early) Results section.

Experimental design

- Methods are reported clearly and concisely, links to available data are provided, all tested tools are specified down to the version number. More Methods sections should look like this! In particular, I would like to commend the authors for uploading all code as reproducible workflow to Github.

- Matthew's Correlation Coefficient (MCC) is used throughout the study as the sole indicator of OTU "quality". The MCC is an objective summary statistic for classification problems and has been successfully used for OTU benchmarking before (originally introduced by the authors in their 2011 paper and since then picked up by several other teams). Nevertheless, I would like to point out three general objections to using the MCC as an OTU quality criterion and then argue that the MCC, when used as a sole index, is not the most informative choice for (all) the questions that the authors pursue in this study.

=> In general, the MCC is a summary statistic for binary classification problems. However, (OTU) clustering is arguably not a problem that boils down to binary true/false assignments only. Of course, violations of a "hard" sequence similarity clustering threshold provide an objective criterion, but different algorithms implement inherently different concepts of a "threshold", and are consequentially prone to different types of errors. For example, by definition, complete linkage or USEARCH would never show "false positive" clustering (but many false negatives can be expected), while single linkage would never make "false negative" mistakes (while a wealth of false positives is expected). In other words, the MCC was not designed for this kind of application and OTU "quality" is not (only) a binary classification problem; nevertheless, MCC values are arguably informative, if interpreted carefully.
=> My second "general" issue with the MCC concerns its formulation as a summary statistic: frankly speaking, I find it rather non-intuitive to interpret. For example, in the present study, it would be very informative to see what actually goes "right" or "wrong" for the different methods, driving MCC values (e.g., by showing distributions of TPR, FPR, etc. per clustering method).
=> Finally, I suspect that the MCC is sensitive to cluster count and size distribution, although I am not aware that this has ever been tested. In general, more "errors" can be expected the further a clustering algorithm proceeds: sets with fewer and larger OTUs would be more prone to erroneous assignments than sets with many small OTUs. This is problematic, because the "hard" clustering threshold is implemented very differently by different algorithms, and it is well established that at a given threshold of, say, 97% the different methods provide very different cluster count and size distributions. In the present study, such effects would be especially striking for the SWARM algorithm.

In spite of these comments, the use of the MCC in the second use case of the present study ("MCC value relative to distances") is arguably established and meaningful, if interpreted cautiously. However, in my opinion, the MCC is an insufficient metric in the first use case ("MCC value relative to full dataset"). The authors generated "reference" OTU sets based on the full datasets and subsequently compared OTU sets based on randomly down-sampled datasets to these using the MCC. In this setup, TPs are sequence pairs which end up in the same OTU under full and down-sampled clustering, etc. Arguably, this corresponds even less to the original use case of the MCC, and putative differential cluster size and count effects would be even more striking (based on the authors' findings in Figure 2). What appears particularly confusing is that the interpretation of the MCC changes drastically between both use cases: the "MCC on distances" is interpretable in line with previous work, but the "MCC relative to full dataset clustering" means something completely different.

Several metrics have been developed which appear more appropriate for the latter use case. The down-sampling experiment, in my opinion, would be more straightforward to interpret if clustering comparison metrics, e.g. based on mutual information were to be used. I freely admit to a "conflict of interest" here: in a previous work I was involved in (DOI: 10.1111/1462-2920.12610), we used Adjusted Mutual Information (Vinh et al., 2010, http://www.jmlr.org/papers/v11/vinh10a.html), Normalized Mutual Information (Fred 7 Jain, 2003, http://ieeexplore.ieee.org/stamp/stamp.jsp?arnumber=1211462) and the Adjusted Rand Index (Hubert & Arabie, 1985, http://link.springer.com/article/10.1007%2FBF01908075) in a very similar setup (testing clustering robustness to selective down-sampling). While I generally consider it very impolite to push one's own work as a reviewer, I do believe that in this case, there is a clear scientific motivation for doing so. For the tests conducted in Fig 1AB & 3AB, at least, I believe that the above measures would provide a highly informative complement, if not replacement, for the MCC. They are straightforward indices of OTU "stability", in the sense defined in the present manuscript. Indeed, based on our previous observations (Fig 5 in Schmidt et al., Environ Microbiol, 2015), I would expect that the trends observed in the present study would be even more pronounced. Taken together, MCC and AMI/NMI/ARI could then indeed be interpreted as OTU "accuracy" and "stability". Likewise, the comparison between USEARCH and VSEARCH is a model use case for which the above-mentioned methods were originally designed.

Validity of the findings

- As detailed above, I believe that all conclusions are generally justified by the analyses conducted, albeit reservations regarding the use of the MCC as sole index in the stated cases. However, as likewise detailed above, I also believe that several findings stand independently of each other, and their connections are not immediately obvious.

- One main rationale of the present study is that OTU assignment "accuracy" trumps "stability". The authors provide clear reasoning to support this argument, and their data is generally in line with this notion. Nevertheless, one could argue that indeed, both "stability" and "accuracy" are important features of OTUs, and that their relative significance depends on the application. It is neither desirable to work with "stable but wrong" nor "accurate but noisy" OTUs. I agree with the authors that accuracy is the more desirable property in general, but putting their study in context with other recent findings could suggest that in fact, accuracy and stability tend to be correlated, which would lend even more gravity to their argument.

- The points raised in critical response to the He et al. study are all well illustrated, argued and supported by data. Moreover, by repeating several of the tests on a more recent (and more reasonably chosen), more deeply sequenced dataset, the authors provide a clear added value.

- The inclusion of tests on the very recent SWARM and VSEARCH algorithms makes this manuscript very timely and useful for researchers in the field. To my knowledge, this is the first available independent test on VSEARCH performance.

- The findings on "problems with reference-based clustering in general and as implemented in QIIME" are striking and, indeed, slightly disturbing. To be honest, I would not have expected such strong effects of sequence order for mappings against the QIIME reference database. I believe that these results will contribute to more debates in the field, and ultimately lead to further long-needed benchmarks on standard operating procedures for widely used software.

- p.12, ll.260-1. This statement is too harsh based on the given data. The SWARM algorithm is designed to not work with "hard" thresholds in the classical sense, so it is not surprising that in the given testing framework, different "thresholds" would appear optimal for differentially deeply sequenced datasets.

- p.15, ll.368-9. This solution appears slightly impractical. While it would clearly be desirable if researchers provided clustering quality metrics along with their results (and ideally, for more than one clustering method), calculating the MCC would always require to calculate and store full sequence distance matrices. Also, it is unclear what an MCC comparison across different studies would actually mean (considering additional confounding factors such as sequencing depth, clustering threshold, alignment, etc.). Nevertheless, I fully agree with the authors that researchers should ideally perform (and report) different clusterings on their data.

Additional comments

I would like to again commend the authors on their manuscript and again apologize for pushing a study their way in which I was involved. I was very hesitant to include the points on the use of the MCC vs AMI/NMI/ARI in this review, but I honestly believe that they make sense scientifically.

---

## Round 0.2 · accepted · Accept

Dear Authors,

Thank you very much for your quick and clear response to the reviewer comments. Both reviewers are now satisfied with your response, and I am sending your submission to the production team.

That being said, the Reviewer 2 (Thomas Schmidt) has some minor, yet important suggestions regarding numerous typos and readability issues in the revised manuscript. Instead of asking for a minor revision, I decided to accept the manuscript in order to speed up the process, and let you address those points during the proof-reading stage. Please consider paying extra attention to these minor issues before sending your proof back to the production team.

Best wishes,

Reviewer 1 ·

Basic reporting

No Comments

Experimental design

No Comments

Validity of the findings

No Comments

·

Basic reporting

See below.

Experimental design

See below.

Validity of the findings

See below.

Additional comments

I commend the authors for their revision of the manuscript and recommend that it be accepted for publication, subject to a few (minor) remaining points.

Figures. The revised figures and figure legends are now (much) more straightforward than in the previous version. However, Fig 5 has a caption, yet the figure seems to be missing: please insert it!

Alternative metrics. As said in my previous review, and as discussed in the e-mails attached by the authors, I would stand by my point that alternative metrics to the MCC could potentially provide a more meaningful assessment of OTU "stability". Nevertheless, the authors have several valid arguments on this, e.g. by pointing out that their primary goal was a critical replication of the He study and the methods used therein. I consider their response to this point completely satisfying.

Methods. The manuscript profited from reorganising the section order. However, the Methods section now starts a little ad hoc: as a first time reader, I would have stumbled over the Roesch analysis, as this is now the first time that the Roesch data is mentioned in the manuscript). I would recommend adding 1-2 sentences to ease the flow for the reader.

Typos. A few typos remain: p.5, l.64, "An alternative"; p.5, ll.83f, "These greedy [...] creates a new index sequence." (possibly use singular form, "create"?); p.12, l.275, "the fewest OTUs"; p.15, l.356 "With the exception of"; p.21, ll.540-1, "has the potential to introduce a systematic bias".

External reviews were received for this submission. These reviews were used by the Editor when they made their decision, and can be downloaded below.